# Improving HIV testing and retention among adolescents and youths: Lessons from a quasi-experimental study of the Red-Carpet Program in Malawi

**Rachel Chamanga** [1] *, **Tessa Musukwa**[1], **Cosima Lenz**[2], **Louiser Kalitera**[1], **Geoffrey Singini**[1], **Felix Gent**[1], **Harrid Nkhoma**[1], **Godfrey Woelk**[2], **Judith Kose**[3,4], **Thulani Maphosa**[1]

**1** Elizabeth Glaser Pediatric AIDS Foundation (EGPAF), Lilongwe, Malawi, **2** Elizabeth Glaser Pediatric AIDS Foundation (EGPAF), Washington, District of Columbia, United States of America, **3** Africa Center for Disease Non-intervention and Prevention (Africa CDC), Nairobi, Kenya, **4** Rotterdam University, Rotterdam, The Netherlands

* rkanyenda@pedaids.org

**Data Availability Statement:** All data are in the manuscript and/or supporting information files.

## Abstract

Adolescents and youth living with HIV (AYLHIV) often face significant challenges in HIV care. Elizabeth Glaser Pediatric AIDS Foundation in Malawi implemented the Red-Carpet Program (RCP) to provide fast-tracked services for AYLHIV in care.This study aimed to assess the effect of RCP on Provider-Initiated HIV testing, linkage to care and antiretroviral therapy (ART), and retention in care among AYLHIV in Blantyre, Malawi. This quasi-experimental study compared outcomes among newly identified AYLHIV enrolled in four intervention health facilities implementing RCP with those of three non-intervention facilities between July 2020 and March 2021. Non-intervention sites were selected by matching based on patient volumes and baseline retention rates prior to the intervention ensuring comparability with the intervention sites. Proportions and Chi-square tests were used to compare outcomes between the two groups. Kaplan-Meier curves were employed to assess longitudinal outcomes, and Cox regression analysis was used to estimate the hazard of non-retention in care. Data were collected from 475 AYLHIV from RCP sites and 248 AYLHIV from non-intervention sites. In the non-intervention sites, 87% of AYLHIV were female, compared to 78% in the RCP sites. A higher proportion of adolescents (67%) underwent provider-initiated HIV testing at intervention site s than at non-intervention sites (51%), p<0.01. Retention in care was higher in RCP sites, with 67% of AYLHIV in care at 12 months post-initiation compared with 56% in non-intervention sites, p = 0.005. AYLHIV from intervention sites were less likely to experience non-retention than those from non-intervention sites (adjusted Hazard Ratio: 0.47, 95% CI: 0.28–0.80). The implementation of the RCP facilitated higher rates of provider-initiated HIV testing among adolescents and youth. Furthermore, RCP demonstrated the potential to improve retention in care The RCP offers promise for enhancing outcomes among this vulnerable population, emphasizing the need for tailored HIV interventions for adolescents and youth.

**Funding:** The authors declare that this work was supported by ViiV Health Care under the Viiv Malawi Grant 519 (Viiv RC 2020-21 – Malawi – 519). The funders had no role in study design, data collection and analysis, decision to publish, or preparation of the manuscript. No additional external funding was received for this study. The authors have no financial relationships or competing interests that could be perceived as influencing the results or interpretation of this research. Its contents are solely the authors' responsibility and do not necessarily represent the official views of Viiv Health Care.

**Competing interests:** The authors have declared that no competing interests exist.

## Introduction

Globally, approximately 2.1 million adolescents and youth navigate the formidable challenges of HIV/AIDS [1]. Within this vulnerable population, adolescents and youth aged 10–24 years living with HIV often have disproportionately inferior outcomes across the HIV care continuum [2]. Compared to adults, they exhibit lower awareness of their HIV status, face substantial hurdles in initiating essential antiretroviral therapy (ART), and exhibit suboptimal viral suppression rates and retention in care [3–5].

The journey of adolescents and youths living with HIV has, in part, been compounded by the historical focus of HIV testing and treatment services on adults, inadvertently neglecting the distinctive requirements of this critical demographic [6]. Adolescence, marked by constant life transitions, experimentation, and engaging in risky behaviours, also face unique factors such as challenges with inconsistent support from caregivers that impact their adherence to ART medication[7]. Non-adherence leads to treatment defaults, virologic failure, and viral rebound, even after initial viral suppression [8–10].

Recognizing the pressing need for tailored, youth-responsive interventions designed to address the unique hurdles confronting adolescents living with HIV, the Elizabeth Glaser Pediatric AIDS Foundation (EGPAF)–Malawi, in collaboration with ViiV Healthcare and the Malawi Ministry of Health, embraced the evidence-based Red-Carpet Program (RCP) that had been conducted by EGPAF in Kenya [11,12]. The RCP's primary focus in Malawi was to enhance HIV testing, linkage to care, and retention in care among adolescents and youth living with HIV, through a youth-centric approach.

Malawi, with its high HIV prevalence rate (8.9%), has witnessed the highest incidence of infections among adolescents and youth aged 15–24 years, with young women being particularly vulnerable due to early sexual debut and entering into relationships with significantly older male partners [13–15]. Paradoxically, despite their heightened risk, young people, especially males, exhibit a reluctance to be tested for HIV [16]. The proportion of those who had never had an HIV test was higher in males aged 15–19 years, with 65% reporting never having an HIV test, compared to 51% in females of the same age group [17]. This gender disparity could be attributed to the traditional male aversion to healthcare-seeking behaviour and the misperception of lower infection risk, unlike females who often access healthcare facilities for reproductive services and testing [16,18,19]. Other influential factors affecting HIV testing engagement include having accurate knowledge about HIV infection, attainment of secondary school education, and household characteristics, such as the educational status of caregivers [16].

Additionally, even among those who undergo HIV testing and are enrolled in care, viral suppression rates remain low, with national estimates as low as 66% [13]. This is in contrast to the 84% suppression rate observed in adults aged 15–49 years in Malawi [13]. Furthermore, retention in care among adolescents and youth ' lags behind that of adults, accentuating the urgency of tailored interventions [20].

The introduction of the RCP in Malawi represents a dedicated endeavour to address the distinctive difficulties faced by adolescents and youth living with HIV, with a special emphasis on leveraging a peer-led approach to identify and retain adolescents and youth in care [11,12]. However, while peer-led initiatives have shown promise in improving HIV testing and ART outcomes, the available evidence remains mixed and limited, especially within the context of sub-Saharan Africa [21–23].

In this context, our evaluation aimed to provide an assessment of the RCP effect on adolescents living with HIV in Blantyre, Malawi. We examined HIV testing outcomes, linkage to ART, and retention in care at 6- and 12-months post ART initiation. This investigation is a

significant step toward improving the understanding of how peer-led initiatives can enhance HIV care outcomes for adolescents and youth, who continue to be disproportionally impacted by the epidemic.

## Methodology

### Study design

This study employed a quasi-experimental program evaluation approach using secondary data collected from adolescents and youth who underwent HIV testing and were newly initiated into ART care between July 2020 and March 2021.

### Study setting

The research was conducted in Blantyre, Malawi, at seven primary health facilities, including four intervention sites participating in the RCP and three non-intervention sites selected based on their similarity in key characteristics to the intervention sites. The characteristics which we matched on between intervention and non-intervention sites included: being a primary care facility in urban settings; having the same volume of newly diagnosed AYLHV and those in HIV care; and having similar baseline retention rates before implementation of the intervention (within ± 5%) We matched on baseline retention rates which are retention rates of the AYLHIV before implementing the program. We matched on this to ensure that there was no bias amongst the facilities with other facilities doing better in retention than others before the intervention.

### Intervention

The Red-Carpet Program, implemented by EGPAF Malawi in collaboration with the Malawi Ministry of Health, introduced a novel approach utilizing Youth Champions (YCs) to support adolescent and youth clients at RCP facilities. YCs, aged 18–25 years, were peer navigators and treatment supporters recruited from among adolescents and youth living with HIV. YCs were volunteers, but received a stipend for their role. Eligibility to become a YC included being fully disclosed to and open to discussing their status and experiences, demonstrating adherence, achieving viral suppression, and having received at a minimum, a secondary school diploma. YCs were strategically stationed at various entry points across the facilities, such as outpatient departments, sexually transmitted infections clinics, antenatal care clinics, HIV testing and screening clinics, family planning clinics, maternity wards, and ART clinics. With the aid of standardized scripts co-developed with young people, YCs engaged with adolescents and youth seeking care at the facility, providing support for HIV testing, counselling, ART initiation, and enrolment in RCP. They also guided those testing negative for risk reduction and prevention support, including pre-exposure prophylaxis. Additionally, YCs offered psychosocial support, adherence support, and retention activities, which encompassed appointment reminders, tracing clients who missed appointments, and conducting home visits for AYLHIV, facing care-related challenges.

### Study population

This study included newly initiated HIV-positive males and females aged 10–24 years who initiated ART between July 2020 and March 2021. At the intervention sites, we included all AYLHIV newly initiated on ART and exposed to the intervention through registration in the Red-Carpet enrolment register; we excluded newly initiated AYLHIV who had not been exposed to the Red Carpet intervention or those newly initiated AYLHIV who had transfer-into the

facility. At the non-intervention sites, all newly initiated AYLHIV on ART, as documented in the ART registry during the same period, were included in the study; we excluded newly initiated AYLHIV who had transferred into the facility.

## Data collection & data analysis

Trained data abstractors collected data from multiple registers, including the HIV testing register, ART register and HIV viral load register. Individual-level data from the electronic medical records (EMR) were also obtained. The data abstractors obtained this data between 20 Oct 21 to 07 Nov 21. The data abstractors obtained de-identified data of the AYLHIV. Data were entered into tablets using the Open Data Kit (ODK) electronic data-collection tool. The abstracted data comprised of age, sex, date of HIV diagnosis, date of ART initiation, ART regimen, date of ART clinic attendance, and HIV treatment outcomes. Treatment outcomes were defined as a) *died*–documented as deceased; b) *alive* and *active in care*—attended the clinic at the last scheduled visit; c) *defaulted*–missed treatment collection or a clinical appointment for 2 or more months after ARVs are expected to have run out with no record of death; d)documented to have *transferred out*–moved to receive services at another facility. Primary outcomes included the proportion of AYLHIV accessing provider-initiated HIV testing (PITC), linkage to care on the same day, and retention in care 6 and 12 months after ART initiation. Quantitative analysis was conducted using STATA v17, involving descriptive analysis, chi-square tests, Kaplan-Meier curves, and Cox regression analysis.

Survival analysis was utilized for two assessments. In the first analysis (Analysis A), we analysed AYLHIV defaulted from care and those who had died as having the outcome of interest, patients who transferred out were treated as intention-to-treat where they contributed their time to the study during analysis but were censored. The second analysis (Analysis B), only that were alive in care were considered to be "retained in care at the facility;" those that had "died", "defaulted" or "transferred out" were considered as having the outcome of interest.

## Ethical considerations

This evaluation was approved by the Malawi National Health Science Research Committee (NHSRC) and the US-based ADVARA Institutional Review Boards under the "Evaluation of Outcomes achieved through integrated HIV/AIDS and TB Prevention, Care and Treatment Programs in Malawi" protocol, which allows for retrospective data review of program data. The protocol was reviewed and approved in accordance with the U.S. Centre for Disease Non-intervention and Prevention (CDC) human research protection procedures, with no direct interaction with human subjects or access to identifiable data or specimens for research purposes. Informed consent or assent was waived by the National Health Research Sciences Committee and ADVARA Institutional Review Board.

## Results

During the study period, 8% (499/6,336) of AYLHIV tested for HIV at RCP sites and 6% (251/4,534) of those tested at control sites tested positive for HIV. Linkage to ART for newly diagnosed AYLHIV was 98% (490/499) at RCP sites compared to 96% (241/251) at control sites. Complete individual data were extracted for 723 newly initiated AYLHIV receiving antiretroviral therapy (ART) (475/499 [95%] AYLHIV from intervention and 248/251 [99%] from non-intervention sites). The total person time for follow up in the study was 627.02 person-days; with 417.14 person-days in the intervention sites and 209.88 person-days in the non-intervention sites.

**Table 1. Demographic and clinical characteristics of adolescents and youths newly initiated into care comparing non-intervention and intervention sites.**

| Variable | Description | Total N = 723 | Non-intervention N = 248 | Intervention N = 475 | P-value |
|---|---|---|---|---|---|
| | | N (%) | N (%) | N (%) | |
| **Sex** | Males | 138 (19.1) | 33 (13.3) | 105 (22.1) | <0.01 |
| | Females pregnant | 208 (28.8) | 58 (23.4) | 150 (31.6) | |
| | Female non-pregnant | 377 (52.1) | 157 (63.3) | 220 (46.3) | |
| **Age at ART initiation** | 10–14 | 27 (3.7) | 5 (2.0) | 22 (4.6) | 0.17 |
| | 15–19 | 168 (23.3) | 62 (25.0) | 106 (22.3) | |
| | 20–24 | 528 (73.0) | 181 (73.0) | 347 (73.1) | |
| **WHO staging** | WHO Stage I/II | 697(96.4) | 242 (97.6) | 455 (95.8) | 0.44 |
| | WHO Stage III/IV | 26(3.6) | 6 (2.4) | 20 (4.2) | |
| **Reason for HIV testing[a]** | Provider-initiated (PITC) | 339 (65.2) | 67 (51.5) | 272 (69.7) | <0.01 |
| | Family referral slip (Active Index Testing) | 122 (23.5) | 48 (36.9) | 74 (19.0) | |
| | Other (Voluntary) | 59 (11.3) | 15(11.6) | 44 (11.3) | |
| **Previous HIV test [a**]** | Negative | 360 (69.2) | 79 (60.8) | 281(72.0) | 0.01 |
| | Positive | 51(9.8) | 11 (8.5) | 40 (10.3) | |
| | Never tested | 108(20.8) | 40 (30.7) | 68 (17.4) | |
| **Initiation on ART[a]** | Newly tested | 468 (90.2) | 119 (91.5) | 350 (89.7) | 0.55 |
| | Previously tested positive | 51 (9.8) | 11 (8.5) | 40 (10.3) | |
| **Initiated ART on same day** | Yes | 697 (96.4) | 236 (95.2) | 461(97.1) | 0.19 |
| | No | 26 (3.6) | 12 (4.8) | 14 (2.9) | |
| **Viral load suppressed[*]** | Yes | 156 (85.7) | 56 (86.2) | 100 (85.5) | 0.89 |
| | No | 26 (14.3) | 9 (13.8) | 17 (14.5) | |

*Those who had viral load results : Intervention sites, n = 117, non-intervention sites n = 65.

** 2 had inconclusive results.

a The numbers do not add to the totals due to missing data. There were 118 missing in non-intervention and 85 missing in intervention sites.

Table 1 provides an overview of the demographic and clinical characteristics of the AYH-LIV included in the study, differentiating between the intervention and non-intervention groups. The overwhelming majority of these AYHLIV cases were female, accounting for 80% of the total cohort; of these female AYLHIV, 35.6% were pregnant. The intervention sites exhibited a higher proportion of males compared to the non-intervention sites (22.1% vs. 13.3%, p<0.01). The age distribution of participants in both groups was not significantly different, with a median age at ART initiation of 21 years (Interquartile Range [IQR] 19–23) across both non-intervention and intervention sites. Notably, all AYLHIV attending both types of sites were initiated on a dolutegravir-based regimen, ensuring consistency in the treatment protocols. The mean time to ART initiation was 0.19 days in intervention sites and 0.25 days in non-intervention sites.

The study revealed a substantial disparity in prior HIV-testing experiences between the intervention and non-intervention sites. A higher proportion of AYLHIV at the intervention facilities had undergone provider-initiated HIV testing than those at non-intervention sites (69.7% vs. 51.5%, p<0.01). Approximately 21% of AYLHIV that were newly initiated on ART had never been tested previously, with a significantly higher proportion observed in the non-intervention sites (30.7% vs. 17.4%, p = 0.01). Furthermore, the data indicated that the majority of newly initiated AYLHIV had tested negative previously, with the intervention sites outperforming the non-intervention sites in this regard by 72.0% and 60.8%, respectively (p = 0.01).

At the time of data collection, all 723 AYLHIV had been on ART for more than six months, rendering them eligible for viral load assessments. However, viral load samples were collected for only a portion of the cohort, with viral load results accessible for only 25.1% (182) of the study population. The overall suppression rate was 86% among this group with viral load testing done. There was no discernible distinction in suppression rates between the non-intervention and intervention sites, with rates of 87% and 86%, respectively (p = 0.89).

At 6-months post-ART initiation, the proportion of AYHLIV retained at the facility was 77.3% in the intervention sites and 68.7% in the non-intervention sites (p-value 0.012) (Table 2). However, these differences became more pronounced at the 12-month interval, with retention dropping to 66.7% at the intervention sites and 56.4% at the non-intervention sites (p-value 0.005). At the end of the first year post-ART initiation, 63.1% of the AYLHIV were successfully retained at the clinic where they had initiated ART;25.0% had transferred to other facilities, 11.6% had defaulted from care, and 0.3% had died.

Breaking down these figures by site type, in the intervention sites, 66.7% of AYLHIV were still actively receiving ART, 12.4% had defaulted, 0.4% had died, and 20.5% had transferred to alternative care settings. In contrast, the non-intervention sites exhibited a slightly lower proportion of AYLHIV who were alive on ART (56.4%), with 10.1% experiencing defaults, 0.0% having died, and 33.5% having transferred out. Notably, approximately 60% of those who transferred out of the study population were non-pregnant women, with 66% of this subgroup falling within the 20–24 years age category.

Analysis of retention rates within intervention and non-intervention sites across various demographic groups including age, sex, and WHO staging revealed interesting patterns. At

**Table 2. Cumulative retention into care at 6 and 12 months by demographic characteristics.**

| Variable | Description | Total | | [a]Retained in care at 6 months | | [a]Retained in care at 12 months | |
|---|---|---|---|---|---|---|---|
| | | Non-intervention | Intervention | Non-intervention | Intervention | Non-intervention | Intervention |
| | | N | N | N (%) | N (%) | N (%) | N (%) |
| **Sex** | Males | 33 | 105 | 23 (69.7) | 80 (76.2) | 21(63.6)* | 64 (60.9) |
| | Females pregnant | 58 | 150 | 44 (75.8) | 124 (82.7) | 41 (70.7) | 109 (72.7) |
| | Female Non-pregnant | 157 | 220 | 104(66.2) | 163 (74.1) | 78 (49.7) | 144 (65.4) |
| **Age at ART start** | 10–14 | 5 | 22 | 5 (100.0) | 17(77.3) | 4(80.0) | 15(68.1) |
| | 15–19 | 62 | 106 | 42 (67.7) | 76(71.7) | 32(51.6) | 61(57.5) |
| | 20–24 | 181 | 347 | 124(67.9) | 274(78.9) | 104(57.4) | 241(69.4) |
| **WHO staging** | WHO Stage I or II | 242 | 455 | 165(68.1) | 350(76.9) | 134(55.4)* | 304(66.8) |
| | WHO Stage III or IV | 6 | 20 | 6(100.0) | 17(85.0) | 6(100.0) | 13(65.0) |
| **Reason for HIV test** | Provider-initiated | 67 | 272 | 47(70.1) | 214(78.6) | 43(64.2) | 181(66.5) |
| | Family referral slip (Active Index Testing) | 48 | 74 | 35(72.9) | 60(81.0) | 29(60.4) | 53(71.6) |
| | Other (Voluntary) | 15 | 44 | 9(60.0) | 32(72.7) | 6(40.0) | 33(75.0) |
| **Previous HIV test** | Negative | 79 | 281 | 59(74.6) | 224(79.7) | 51(64.5) | 200(71.2) |
| | Positive | 11 | 40 | 5(45.4) | 25(62.5) | 5(45.4) | 20(50.0) |
| | Never tested | 40 | 68 | 27(67.5) | 56(82.3) | 22(55.0) | 46(67.6) |
| **Method into ART** | Newly tested | 119 | 350 | 86(72.9) | 281(80.2)* | 73(61.3) | 247(70.5)* |
| | Previously tested positive | 11 | 40 | 5(45.4) | 25(62.5) | 5(45.4) | 20(50.0) |
| **Viral load suppressed** | Yes | 57 | 100 | 57(100.0)* | 100(100.0) | 52(91.2) | 95(95.0) |
| | No | 9 | 17 | 8(88.9) | 17(100.0) | 8(88.9) | 16(94.1) |
| Total | | 248 | 475 | 171(68.7) | 367) (77.3) | 140 (56.4) | 317 (66.7) |

[a] Retained into care were only those who were "Alive and on ART", this excluded deaths, transfer outs and defaulters. *p value of Chi-square test was $\leq$ 0.05.

the intervention sites, insignificant differences were observed in retention rates across the age, sex, and WHO staging. However, significant variations in retention rates were observed within the non-intervention sites: Females who were not pregnant exhibited the lowest retention rate at 49.7%, in contrast to males (63.6%) and pregnant females (70.7%) at the non-intervention sites at the 12-month mark (p = 0.01). In contrast, in the intervention sites, the proportion of females who were not pregnant and retained into care was notably higher, hovering around 65.4%, with males having the lowest proportion of being retained into care at 60.9% and pregnant females having the highest proportion at 72.7%; the differences between males and females were not significant (p = 0.13).

Furthermore, older adolescents aged 15–19 years displayed a lower proportion of retention in care compared to other age groups (10–14 and 20–24 years) at the 12-month juncture, although this difference did not reach statistical significance [(57.5% vs. 68.1% vs. 69.4%; p = 0.07 in intervention sites) (51.6% vs. 80.0% vs. 57.4% in non-intervention sites, p = 0.41)]. However, those with advanced WHO stage III or IV experienced higher proportions of retention than those with WHO stage I or II at the 12-month mark in the non-intervention sites (100.0% vs. 55.1%; p = 0.03), while this difference was less pronounced in the intervention sites (65.0% vs. 66.8%, p = 0.86).

Adolescents and youth who had previously tested positive for HIV exhibited lower proportions of active care engagement than those who were newly diagnosed at the 12-month mark in intervention sites (50.0% vs. 70.5%; p<0.01). A similar trend was noted in the non-intervention sites, where 45.4% of previously diagnosed individuals were in care compared to 61.3%, although this difference was not statistically significant (p = 0.30). Moreover, a lower proportion of AYLHIV with an unsuppressed viral load was engaged in active care compared to those with suppressed viral loads at the 6-month point in the non-intervention sites (88.9% vs. 100.0%; p = 0.01).

In Fig 1A and 1B, Kaplan-Meier curves illustrate the comparative retention outcomes between the intervention and non-intervention groups stratified by whether patient transfer is censored or included as outcome. Analysis A (when patients who transfer out are censored at time of transfer), no appreciable difference in retention was observed when comparing the non-intervention and intervention sites, as evidenced by a log-rank test p-value of 0.85. However, in Analysis B (when those who transferred out are included as outcome of interest), retention is significantly different between the non-intervention and intervention sites, with a log-rank test p-value of 0.005.

Table 3 further illuminates the differences in hazard ratios of the risk of being lost to care between non-intervention and intervention sites, adjusting for age and sex. In Analysis A, there is no significant difference in the hazard of being lost in care, with an adjusted hazard ratio (aHR) of 0.95 (95% confidence interval [CI]: 0.37–2.42).However, Analysis B shows a substantially decreased hazard of being lost to care in the intervention sites when compared to the non-intervention sites, as evidenced by an aHR of 0.47 (95% CI: 0.28–0.80).

## Discussion

In this quasi-experimental evaluation of the Red-Carpet Program in Malawi, we demonstrated significant improvements in Provider Initiated Testing and Counselling (PITC) services, specifically tailored to adolescents and youth in the intervention sites compared to the non-intervention sites. These findings align with other studies conducted in Kenya, which have shown enhanced HIV-testing outcomes among adolescents and youth in similar program settings [11].

The Red Carpet Program strategically addresses critical issues related to implementing adolescent-friendly services, fostering increased screening and testing among eligible adolescents

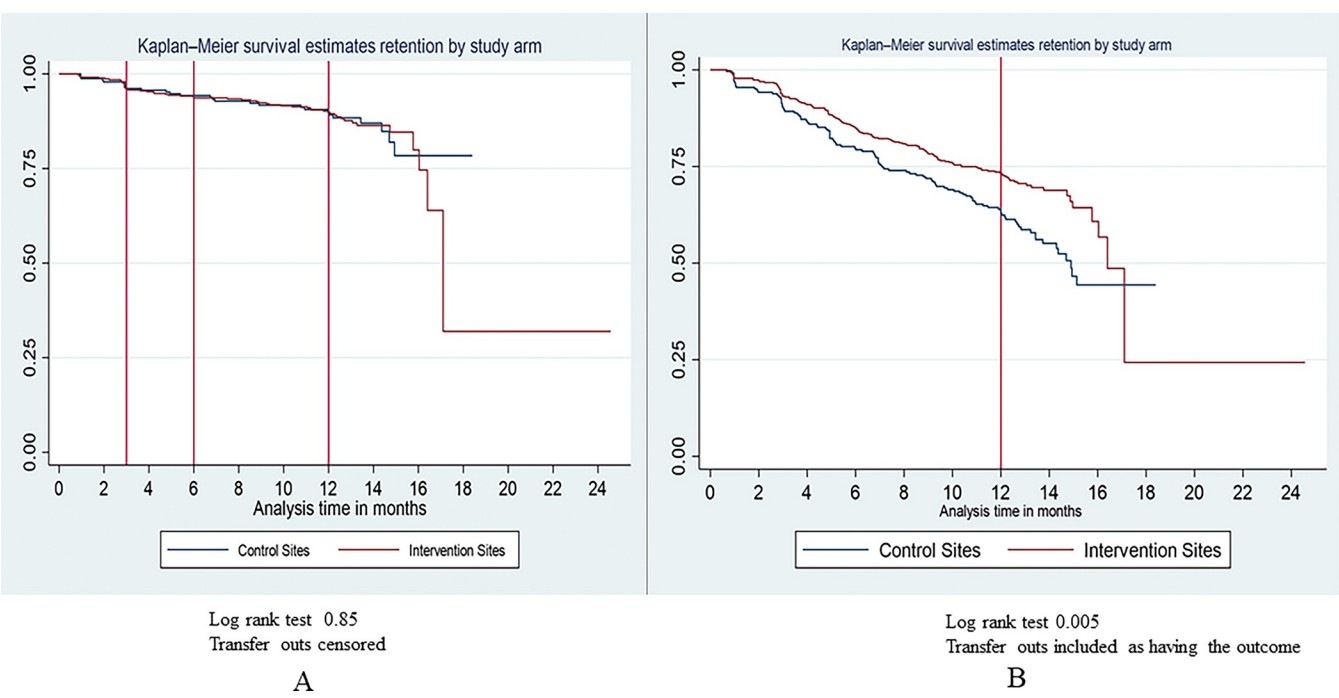

**Fig 1. Kaplan Meier curve describing retention in care for non-intervention vs intervention sites.**

and youth seeking care at health facilities [12]. This was evidenced by the identification of more male adolescents living with HIV, and a reduction in the proportion of adolescents and youths who had never undergone HIV testing in intervention sites. This trend echoes the positive outcomes observed in a study conducted in Malawi, where the establishment of adolescent-friendly clinics, known as "Girl Power" clinics, led to a 23% increase in HIV testing among adolescents and young women when compared to the standard of care [24].

Moreover, similar increases in HIV testing willingness and uptake among adolescents and youth have been reported in other settings utilizing peer-interventions, as reported in a study in Ethiopia that assessed peer-led HIV behavior education in schools, which resulted in an increase in knowledge and increased interest in seeking HIV testing among students in the intervention group [21]. However, the fidelity of implementing adolescent-friendly services remains a concern, with studies showing that only 40% of expected facilities provide such services [25]. Ensuring the widespread availability and accessibility of youth-responsive services in all health facilities is a challenge that adolescent programs continue to face [25].

**Table 3. Cox Proportional Hazard of lost to care; non-intervention and intervention sites.**

| | | Analysis A | | Analysis B | |
|---|---|---|---|---|---|
| **Variable** | **Description** | **Crude (95% CI)** | **Adjusted (95% CI)** | **Crude (95% CI)** | **Adjusted (95% CI)** |
| **Intervention** | Non-intervention | 1.00 | 1.0 | 1.0 | 1.0 |
| | Intervention | 1.09 (0.43–2.72) | 0.95(0.37–2.42) | 0.48 (0.28–0.81) | 0.47 (0.28–0.80) |
| **Sex** | Females | 1.00 | 1.0 | 1.0 | 1.0 |
| | Males | 1.88 (0.76–4.67) | 1.86(0.72-4-79) | 1.01 (0.53–1.92) | 1.22 (0.63–2.33) |
| **Age at ART start** | 10–14 | 1.00 | 1.0 | 1.0 | 1.0 |
| | 15–19 | 0.45 (0.05–4.39) | 0.57 (0.06–5.79) | 2.42 (0.31–18.45) | 2.29 (0.29–17.7) |
| | 20–24 | 0.71 (0.09–5.41) | 0.89(0.11–7.06) | 2.06 (0.28–15.02) | 1.99 (0.27–14.7) |

Despite promising advancements in HIV testing, the overall retention rates at 12 months post-ART initiation (63.1%) are lower than those reported in other studies [26–28]. For instance, a study in South Africa documented retention rates of 90.5% and 85.4% 6- and 12-months after post-ART initiation, respectively [27]. Similarly, a cohort study in Namibia reported 6- and 12-month retention rates of 97.7% and 94%, respectively [28]. In contrast, lower retention rates of 49% at 1-year post-ART initiation were reported in Mozambique [8], and Uganda observed a 65% retention rate among adolescents in care after 5 years of follow-up [29]. These discrepancies may be attributed to various factors, including the local context and rigor of program implementation.

A nuanced examination revealed higher proportions of AYLHIV who were alive and receiving ART at intervention facilities than at non-intervention sites at both 6 and 12 months, specifically when calculated using cumulative retention. However, these differences were not apparent when the transfer-outs were censored in the survival analysis. This discrepancy in outcomes aligns with findings from other studies that reported variations in retention outcomes depending on whether transfer-outs were included or excluded from the analysis [30]. Our study had a relatively higher proportion of participants classified as transfer-outs (25%), which exceeds the percentages reported in other studies, which tend to be approximately 16% [20,31]. Most of these transfers occurred in non-intervention sites, which could have been because the RCP intervention reduced patient desire to transfer out. Interventions that are focused on adolescents in Malawi have been shown to provide greater retention into care [32]. The transfers mirrored a prevalent pattern where female adolescents and young adults, particularly those who are not pregnant, exhibit a higher likelihood of being lost to care or transferring due to their inherent mobility and potential migration (for example, a non-pregnant female AYLHIV who gets married may move to her spouses location) [31,33].This was however in contrast with findings from South Africa, where pregnant adolescents and young females demonstrated lower retention rates than their non-pregnant counterparts [26]. Similarly, a study from Malawi focusing on women receiving Option B+ treatment reported higher rates of attrition among pregnant or breastfeeding women than among those who were not pregnant. However further research is required to understand the reasons why transfer-outs are more prevalent in AYLHIV in Malawi.

Furthermore, our study identified that adolescents and youths who had previously tested positive for HIV but had not yet started ART exhibited lower retention than those newly diagnosed in the non-intervention sites. This delay in ART initiation could be attributed to factors such as denial of HIV status, fear of stigma, lack of disclosure of HIV status, concerns about side effects, long waiting times at ART clinics, and transportation challenges [33]. These issues continue to influence adolescents and youth and affect their retention of care.

Finally, our study revealed that AYLHIV in WHO Clinical Stage III or IV had higher retention rates than those in WHO Clinical Stage I or II at non-intervention sites. This aligns with the findings of other studies, in which advanced WHO staging was not associated with attrition [29]. Those with advanced HIV disease may be more motivated to seek care because of their health condition, which leads to closer follow-up and better retention [26,29].

This study has some limitations, including gaps in updating electronic medical records (EMRs) which could lead to delays in reported outcomes. The study also utilised a quasi-experimental design where the selection of control and intervention sites was not randomised and thus it is difficult strongly make causal associations. We also incurred missing data for some of the key variables such as viral load which may have affected calculation of the outcomes.

## Conclusion

In conclusion, our study highlights the impact of the RCP in increasing provider-initiated HIV testing and enhancing retention rates among adolescents and youth in Malawi. However,

while retention was better at intervention sites, retention at care at 12-months post ART initiation was not optimal in either group. Notably, this study revealed the vulnerability for poor retention of non-pregnant females, those in WHO stage I or II, and individuals who previously tested positive for HIV but did not initiate ART, underlining the pressing need for tailored interventions. These findings underscore the importance of ongoing efforts to deliver effective adolescent-friendly services, promote early ART initiation, and address the unique challenges faced by adolescents in their fight against HIV/AIDS.

## Supporting information

**S1 Data. Study data set.**
(XLSX)

## Author Contributions

**Conceptualization:** Rachel Chamanga, Tessa Musukwa, Cosima Lenz, Godfrey Woelk, Judith Kose, Thulani Maphosa.

**Formal analysis:** Rachel Chamanga, Cosima Lenz, Louiser Kalitera, Geoffrey Singini, Harrid Nkhoma.

**Funding acquisition:** Cosima Lenz.

**Methodology:** Rachel Chamanga, Tessa Musukwa, Cosima Lenz, Louiser Kalitera, Godfrey Woelk, Judith Kose, Thulani Maphosa.

**Supervision:** Godfrey Woelk, Judith Kose, Thulani Maphosa.

**Writing – original draft:** Rachel Chamanga, Tessa Musukwa, Cosima Lenz, Geoffrey Singini, Felix Gent, Godfrey Woelk, Judith Kose, Thulani Maphosa.

**Writing – review & editing:** Rachel Chamanga, Cosima Lenz, Louiser Kalitera, Felix Gent, Godfrey Woelk, Thulani Maphosa.

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
