## [Decision Letter · Decision Letter 0]

19 Jun 2024

PGPH-D-24-00709

Improving HIV Testing and Retention among Adolescents: Lessons from the Red-Carpet Program in Malawi

Dear Dr. Chamanga,

Thank you for submitting your manuscript to PLOS Global Public Health. After careful consideration, we feel that it has merit but does not fully meet PLOS Global Public Health’s publication criteria as it currently stands. Therefore, we invite you to submit a revised version of the manuscript that addresses the points raised during the review process.

EDITOR:

Overall the manuscript is well written. However, I request you to kindly address the following-

1. The manuscript refers to both adolescents and young adults, hence please consider adding ' in the title. 

2. The word 'impact' is used in the last paragraph of the introduction. Kindly replace with effect as impact is a long term effect and this study assesses outcomes at 6 and 12 months post ART initiation.

Kindly clarify if all the AYLHIV in records at the intervention and nonintervention sites were included. If all were included, a sample size calculation is not necessary.

3. It is unclear what 'baseline retention rates' in the inclusion of sites means. 

Also non-intervention sites were selected based on numbers of adolescents in care and retention rates at would it not affect the outcome studied? Selecting sites based on similar rates of diagnosis would be more appropriate. 

4. Were all AYLHIV during the study period at each site included? If not what were the exclusion criteria e.g., incomplete records, those diagnosed at the facility but belonging to a different region.

5. Were AYLHIV who missed the last visit </=2 months ago included in alive and active in care? Usually what is the practice regarding dispensing medications in Malawi?- is this monthly or once in 2 months?

6. Is there a better term for Defaulted? Health programmes tend to use the tearm lost-to-foloow-up instead of default. Kindly consider changing this- unless ofcourse the National Program in Malawi uses the tearm 'default'. Also, does default include AYLHIV who returned to care with more than 2 months of missing follow-ups?

7. Please indicate the total person-time follow-up in the study and in the intervention and non-intervention arms. Also, would be good to have mean/median time to treatment initiation.

8. Kindly discuss limitations of the methodology in a bit more detail- the quasiexperimental design, the selection of nonintervention sites, quality of records and the data at all sites to name a few. On the other hand, small sample size is generally not included under limitations, especially, if all records were included. Reasons why all records were not included could be a limitation.

9. Minor language edits necessary e.g., "AYLHIV patients", 63.1% of the "patients with AYLHIV" and "the overall retention rates at 12 months after post-ART".  Also kindly use standard teminilogy through-out the document,

We look forward to receiving your revised manuscript.

Kind regards,

Rashmi Josephine Rodrigues, M.D., Ph.D.

Academic Editor

Journal Requirements:

2. Please provide separate figure files in .tif or .eps format only and remove any figures embedded in your manuscript file. Please also ensure all files are under our size limit of 10MB.

3. In the online submission form, you indicated that "The data set is readily available on request by the journal". 

3. Uploaded as supplementary information.

Additional Editor Comments (if provided):

Reviewers' comments:

Reviewer's Responses to Questions

**Comments to the Author**

1. Does this manuscript meet PLOS Global Public Health’s publication criteria? Is the manuscript technically sound, and do the data support the conclusions? The manuscript must describe methodologically and ethically rigorous research with conclusions that are appropriately drawn based on the data presented.

Reviewer #1: Yes

Reviewer #2: Yes

2. Has the statistical analysis been performed appropriately and rigorously?

Reviewer #1: Yes

Reviewer #2: Yes

3. Have the authors made all data underlying the findings in their manuscript fully available (please refer to the Data Availability Statement at the start of the manuscript PDF file)?

Reviewer #1: Yes

Reviewer #2: Yes

4. Is the manuscript presented in an intelligible fashion and written in standard English?

Reviewer #1: Yes

Reviewer #2: Yes

5. Review Comments to the Author

Reviewer #1: While their research methodology is robust and their findings significant, the article requires minor corrections. Overall, this study serves as a crucial contribution to the field, highlighting the urgent need for action in addressing the strategies to improve HIV screening and retention in care to achieve the 95*95*95 targget by 2025. Thank you for the opportunity to review this enlightening article.

Reviewer #2: At the outset, I must congratulate the authors of the study and the EGPAF team at Malawi for the wonderful work you all are doing at the helm of the HIV epidemic. I must definitely appreciate the efforts taking in improving the AYLHIV outcomes, a very important subgroup of the PLHIV who require extra care and effort which the RCP is an embodiment of.

Strengths of the Study:

1. Well structured study with good methodological explanation.

2. An important separation with use of two groups for Analysis based on the transferred out individuals which could act as a large confounding group in evaluation of your study.

3. Intelligently written and well presented study.

Suggestions:

I had a few semantic errors which I have brought to your notice:

Line 52 - Would suggest use of p =0.012, instead of just 0.012.

Line 148 - Would suggest use of d) for the 4th treatment outcome " transferred out"

Table 1 : How has the missing data been analysed? included or not included in analysis?

Line 219 - Would suggest WHO Staging instead of WHO Stagingg.

Recommendations:

1. Would consider further research and analysis into the transferred out, to understand the underlying factors which contribute to the same since it is a large percent of the cohort which was censored.

2. Would advocate for the low viral load testing done in your participants as it has significantly reduced the numbers and hence the outcome. To consider ways to boost viral load testing more robustly with use of YCs or an intervention which is deemed fit by you all.

3. The full title and short title uses keywords like adolescents which by WHO definition would be just 10-19years, but the median age of ART initiation was 21 years which is more in the bracket of young adult. However there are multiple papers which believe comprehensive adolescent care should include upto 24 years and hence the same must be highlighted in your paper as a need for inclusion of the age group. I think it is important to reflect the same in the title like use of AYLHIV in the title.

6. PLOS authors have the option to publish the peer review history of their article (what does this mean?). If published, this will include your full peer review and any attached files.

**Do you want your identity to be public for this peer review?** For information about this choice, including consent withdrawal, please see our Privacy Policy.

Reviewer #1: **Yes: **Claudia Merlin Tony

Reviewer #2: **Yes: **Nikith Austin D'Souza

---

## [Decision Letter · Decision Letter 1]

28 Oct 2024

PGPH-D-24-00709R1

Improving HIV Testing and Retention among Adolescents and Youth: Lessons from the quasi-experimental study of the Red-Carpet Program in Malawi

Dear Dr. Chamanga,

Thank you for submitting your manuscript to PLOS Global Public Health. After careful consideration, we feel that it has merit but does not fully meet PLOS Global Public Health’s publication criteria as it currently stands. Therefore, we invite you to submit a revised version of the manuscript that addresses the points raised during the review process.

Kindly address the comments by the reviewer in the attached documents as well as the sanitised copy of the manuscript.

We look forward to receiving your revised manuscript.

Kind regards,

Rashmi Josephine Rodrigues, M.D., Ph.D.

Academic Editor

Journal Requirements:

Additional Editor Comments (if provided):

Reviewers' comments:

Reviewer's Responses to Questions

**Comments to the Author**

1. If the authors have adequately addressed your comments raised in a previous round of review and you feel that this manuscript is now acceptable for publication, you may indicate that here to bypass the “Comments to the Author” section, enter your conflict of interest statement in the “Confidential to Editor” section, and submit your "Accept" recommendation.

Reviewer #1: (No Response)

Reviewer #2: All comments have been addressed

2. Does this manuscript meet PLOS Global Public Health’s publication criteria? Is the manuscript technically sound, and do the data support the conclusions? The manuscript must describe methodologically and ethically rigorous research with conclusions that are appropriately drawn based on the data presented.

Reviewer #1: Yes

Reviewer #2: Yes

3. Has the statistical analysis been performed appropriately and rigorously?

Reviewer #1: Yes

Reviewer #2: Yes

4. Have the authors made all data underlying the findings in their manuscript fully available (please refer to the Data Availability Statement at the start of the manuscript PDF file)?

Reviewer #1: Yes

Reviewer #2: Yes

5. Is the manuscript presented in an intelligible fashion and written in standard English?

Reviewer #1: Yes

Reviewer #2: Yes

6. Review Comments to the Author

Reviewer #1: (No Response)

Reviewer #2: Queries have been addressed adequately!

7. PLOS authors have the option to publish the peer review history of their article (what does this mean?). If published, this will include your full peer review and any attached files.

**Do you want your identity to be public for this peer review?** For information about this choice, including consent withdrawal, please see our Privacy Policy.

Reviewer #1: No

Reviewer #2: **Yes: **Nikith Austin DSouza

---

## [Decision Letter · Decision Letter 2]

28 Nov 2024

Improving HIV Testing and Retention among Adolescents and Youth: Lessons from the quasi-experimental study of the Red-Carpet Program in Malawi

PGPH-D-24-00709R2

Dear Dr Chamanga,

We are pleased to inform you that your manuscript 'Improving HIV Testing and Retention among Adolescents and Youth: Lessons from the quasi-experimental study of the Red-Carpet Program in Malawi' has been provisionally accepted for publication in PLOS Global Public Health.

Best regards,

Rashmi Josephine Rodrigues, M.D., Ph.D.

Academic Editor

Reviewer Comments (if any, and for reference):

Reviewer's Responses to Questions

**Comments to the Author**

1. If the authors have adequately addressed your comments raised in a previous round of review and you feel that this manuscript is now acceptable for publication, you may indicate that here to bypass the “Comments to the Author” section, enter your conflict of interest statement in the “Confidential to Editor” section, and submit your "Accept" recommendation.

Reviewer #1: All comments have been addressed

2. Does this manuscript meet PLOS Global Public Health’s publication criteria? Is the manuscript technically sound, and do the data support the conclusions? The manuscript must describe methodologically and ethically rigorous research with conclusions that are appropriately drawn based on the data presented.

Reviewer #1: Yes

3. Has the statistical analysis been performed appropriately and rigorously?

Reviewer #1: Yes

4. Have the authors made all data underlying the findings in their manuscript fully available (please refer to the Data Availability Statement at the start of the manuscript PDF file)?

Reviewer #1: Yes

5. Is the manuscript presented in an intelligible fashion and written in standard English?

Reviewer #1: Yes

6. Review Comments to the Author

Reviewer #1: Kind Madam/Sir,

I hope this message finds you well. I have thoroughly reviewed your article titled " Improving HIV Testing and Retention among Adolescents and Youth: Lessons from the quasi- experimental study of the Red-Carpet Program in Malawi" and wanted to share my feedback with you.

Strengths: The appropriate changes are made as suggested. Overall, your article presents a valuable contribution to HIV management. The thorough analysis and the innovative approach to methodology are commendable. Your insights have the potential to significantly influence future research in this area.

Thank you once again for the opportunity to review your work. All the best on your future endeavors.

Best regards

7. PLOS authors have the option to publish the peer review history of their article (what does this mean?). If published, this will include your full peer review and any attached files.

**Do you want your identity to be public for this peer review?** For information about this choice, including consent withdrawal, please see our Privacy Policy.

Reviewer #1: **Yes: **Claudia Merlin
